# Numerical Study on the Characteristics of Methane Hedging Combustion in a Heat Cycle Porous Media Burner

**Fei Wang** [1,2], **Xueming Li** [1,*], **Shuai Feng** [3] **and Yunfei Yan** [3,*]

1   College of Chemistry and Chemical Engineering, Chongqing University, Chongqing 401331, China; wf20023184@163.com
2   Chongqing Special Equipment Inspection and Research Institute, Chongqing 401121, China
3   Key Laboratory of Low-Grade Energy Utilization Technologies and Systems, Chongqing University, Ministry of Education, Chongqing 400030, China; shuaifengcqu@cqut.edu.cn
*   Correspondence: xuemingli@cqu.edu.cn (X.L.); yunfeiyan@cqu.edu.cn (Y.Y.)

**Abstract:** With the rapid development of portable devices and micro-small sensors, the demand for small-scale power supplies and high-energy-density energy supply systems is increasing. Comparing with the current popular lithium batteries, micro-scale burners based on micro-thermal photoelectric systems have features of high power density and high energy density, the micro-scale burner is the most critical part of the micro-thermal photovoltaic system. In this paper, the combustor was designed as a heat cycle structure and filled with porous media to improve the combustion characteristics of the micro combustor. In addition, the influence of the porous media distribution on the burner center temperature and wall temperature distribution were studied through numerical simulation. Furthermore, the temperature distribution of the combustor was studied by changing the porous media parameters and the wall parameters. The research results show that the heat cycle structure can reduce heat loss and improve combustion efficiency. When the combustion chamber is filled with porous media, it makes the radial center temperature rise by about 50 K and the temperature distribution more uniform. When filling the heat cycle channel with porous media the wall temperature can be increased. Finally, the study also found that as methane is combusted in the combustor, the temperature of the outer wall gradually increases as the intake air velocity increases. The results of this study provide a theoretical and practical basis for the further design of high-efficiency combustion micro-scale burners in the future.

**Keywords:** heat cycle; porous media; temperature distribution; wall parameters

## 1. Introduction

With the rapid development of portable devices and micro-sized sensors, there is an increasing demand for small-scale power supplies and high-energy-density energy supply systems. Comparing with the current popular lithium battery products on the market, micro-scale burners based on hydrocarbon fuels have characteristics of high power density and high energy density [1–6]. With the advancement of science and technology, the micro-motor system has the advantages of a high power density and a long energy supply time, providing sufficient power for micro-equipment. However, there are problems such as friction, fabrication, and installation of the current micro power generation devices. In the micro-thermal photoelectric system, there are no moving parts, and the device can be used to convert the radiant energy on the outer wall of the burner into electrical energy. The micro-burner is the main device of the micro-thermal photovoltaic system. For the micro-thermal photovoltaic system, its energy conversion efficiency is affected by the performance of the micro-burner [7–9]. Low-calorific-value energy, such as methane has low utilization rate, low combustion efficiency, and excessive pollutant emissions. The development of technologies that can efficiently and stably utilize methane can effectively alleviate existing energy problems. Porous media combustion technology has advantages of high

combustion efficiency, such as low heat loss, and stable combustion, which can effectively solve the problem of low energy efficiency with a low calorific value. The regenerative porous medium hedge burner has superior characteristics, that is, uniform temperature distribution, less heat loss, high combustion efficiency, and stable combustion. It can be applied well to micro-thermal photovoltaic systems to improve the energy conversion rate and has a wide range of application prospects and important research significance.

In recent years, researchers have continued to study, design, and perfect micro-burners. However, with the reduction of the size of the burner, an increase of combustion heat loss and enhancement of combustion instability occur. The key problem of combustion technology urgently needs to be solved. At present, the basic methods to improve the combustion characteristics of micro-burners mainly include porous media combustion technology, regenerative combustion technology, and catalytic combustion technology [10–13]. Adopting porous medium combustion technology, the heat generated by the combustion in the micro burner can fully preheat the unburned gas with the good thermal conductivity of the porous medium, so that the combustion in the combustion chamber is sufficient, the combustion efficiency of the micro burner is improved, and the heat loss is reduced. The combustion is more stable. The use of regenerative combustion technology can use the heat of the combustion tail gas to preheat the unburned gas in the regenerative channel, which increases the combustion efficiency and reduces the heat loss. In recent years, many researchers have studied the laws of combustion characteristics in the micro-burner, considering the influence of various factors in the burner on the combustion, and have achieved good results. Coutts [14] et al. solved the difficulties in the application of thermo-optical technology by studying thermo-optical systems. White et al. [15] designed a micro-thermal photoelectric system, which is composed of a micro-burner and a photovoltaic cell. It was found that the maximum energy conversion efficiency of the system was 13%, which is higher than the efficiency of the earliest thermal photoelectric system. Mao et al. [16] numerically simulated the output curve of a micro-thermal photovoltaic system composed of a silicon carbide radiator and a silicon cell. The calculation results showed that as the temperature of the silicon carbide radiator decreases, the output power of the system decreases rapidly, and the efficiency of the photovoltaic cell gradually decreases, the opposite is true when the temperature rises.

In addition, various micro-scale combustion technologies have continuously appeared, and porous media combustion technology and regenerative combustion technology can solve well the problems caused by the reduction of combustion size [17,18]. The basic principle of regenerative combustion is that after the fuel gas is burned in the combustion chamber, the exhaust gas produced after combustion still has a lot of heat. The exhaust gas preheats the new fuel gas through the regenerative channel, increasing the enthalpy of the unburned fuel, thereby increasing the combustion rate, which makes the combustion more complete. Regenerative combustion can use the preheating of exhaust gas to improve combustion efficiency and combustion rate, reduce heat loss, and enhance combustion stability. Ronny et al. [19] invented a micro burner that has no moving parts and is simple to make and install. It is called a "Swiss roll". The burner has a large convective heat exchange area, and the heat released by the combustion in the combustion chamber can heat the unreacted gas through the regenerative wall surface, which greatly improves the energy utilization rate of the burner, and fully reflects the regenerative combustion advantage. The regenerative combustor invented by Cao et al. [20–22] tested the performance of the combustor under different operating conditions. The results showed that this new type of regenerative structure can prolong the fuel staying in the combustor. The combustion is more complete, and the unburned gas can be preheated well, so that the combustion efficiency of the burner is improved. Donoso et al. [23] studied the combustion characteristics in the porous media regenerative combustor through experimental simulation and determined the relationship between the burner parameters and the combustion characteristics. The research results showed that the combustion limit of the combustor is wider when the diffusion combustion is used. Porous media burners can maintain good

performance even with high excess air coefficient and high combustion power. Ma et al. [24] added porous media to the Swiss roll. The Swiss roll structure can make good use of the heat of the combustion exhaust gas to preheat the unburned gas, reduce the heat loss of combustion, increase the combustion reaction rate, and improve the combustion efficiency as well as expand the combustion limit. At the same time, because the porous medium has good heat conduction and heat storage effects, the temperature distribution during the combustion of the porous medium is more uniform. At present, regenerative burners have been researched in depth. Regenerative burners can make good use of waste heat to heat unburned gas, which not only saves energy, but also improves combustion efficiency, which can effectively solve the problem of micro-scale burners. Therefore, the research on regenerative burners has important significance and good prospects. Hedging combustion means that two fuels are injected into the combustion chamber, collide from opposite directions, and burn in the combustion chamber. The hedging combustion can increase the disturbance and make the combustion more violent and complete. At present, domestic scholars are mainly studying the characteristics of hedging flames. Barra et al. [25] used numerical simulation to study the combustion process of methane premixed gas in porous media and studied the preheating characteristics of the unburned methane premixed gas by porous media. The reduction of the equivalence ratio of the mixed gas increases, and the thermal conductivity and radiation emissivity of the porous medium have a great influence on the heat conduction process. In order to improve the micro burner, Li et al. [26] filled the micro burner with porous media. The porous media material was silicon carbide with a porosity of 0.85. The research results showed that when the micro burner was filled with porous media, its wall temperature compared with the non-porous medium, had an increase of about 100 K, and the filling of the porous medium also extended the combustion limit of the micro combustor. They also studied the relationship between the intake air flow rate and the temperature of the outer wall of the micro-combustion chamber.

With the development of the micro-thermal photoelectric system, the requirements for the combustion stability of the micro-burner are getting higher, while the advantages and disadvantages of the micro-burner affect the efficiency of the entire system. Adopting a heat recycle structure can improve combustion stability and reduce heat loss while filling porous media can enhance combustion efficiency. In this paper, the combustor was designed as a heat cycle structure for the micro-thermal photovoltaic system, which adopted the premixed gas hedging combustion method. In addition, $Al_2O_3$ ceramic foam was selected as the porous medium material, and its influence on the temperature distribution of the burner was studied. Furthermore, the relationship between the regenerative porous media combustor at different intake air flow rates, equivalence ratio, the axial temperature of the combustor, and the temperature distribution of the outer wall surface were investigated for insight into the factors affecting the performance of the combustor to obtain the best performance of the combustor.

## 2. Physical Mathematical Model

### 2.1. Physical Model

The physical model used in this paper is shown in Figure 1. The heat cycle burner has a cylindrical structure. The total length of the heat cycle burner is 12 mm, the fuel inlet diameter $d_1$ = 1 mm, the outer diameter $d_2$ = 3 mm, the diameter of the combustion chamber $d_3$ = 4 mm, the diameter of the burner $d_4$ = 5.6 mm, and the partition thickness I = 0.3 mm. The combustor is provided with three sections, namely the air intake passage, the reaction section, and the heat cycle channel. The air intake section and the heat cycle channel are separated by a baffle. The premixed fuel gas enters the combustion chamber from the intake passages on both sides, is opposed in the combustion chamber, and then ignited and burned. The combusted flue gas enters the heat cycle passage from the outlets on both sides, and the exhaust gas preheats the unburned gas, which improves the efficiency and stability of combustion. Porous medium is added to the combustion chamber. The

material of the porous medium is $Al_2O_3$. The addition of porous medium can make both the temperature distribution and the flame in the combustion chamber more uniform.

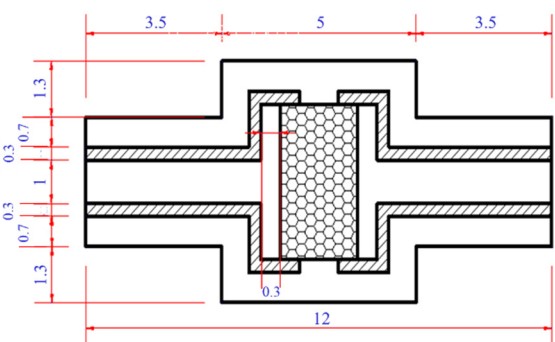

**Figure 1.** Physical model of the heat cycle burner structure.

### 2.2. Mathematical Model

In the simulation process of this paper, it is assumed that the mixture of fuel and air has been fully mixed before entering the combustor, ignoring the radiation effect and viscous dissipation effect of the gas. At the same time, it is assumed that the porous medium is a homogeneous gray body of the same nature. The catalytic effect of the solid high temperature is ignored, and the solid wall and the partition have no slip. Considering the heat exchange between the wall and the environment, the partition and the inner wall are both radiant gray bodies. Under the influence of gravity, the zero physical quantity gradient is located at the cross section of the cylinder, and the premixed gas of methane and air is regarded as an incompressible ideal gas. The relevant control equations involved are as follows:

Continuity equation:

$$\frac{\partial(\varepsilon\rho U)}{\partial x} + \frac{\partial(\varepsilon\rho V)}{\partial y} = 0 \tag{1}$$

Momentum equation:

$$\frac{\partial(\varepsilon\rho UU)}{\partial x} + \frac{\partial(\varepsilon\rho UV)}{\partial y} = -\frac{\partial P}{\partial x} + \frac{\partial}{\partial x}(\mu\frac{\partial U}{\partial x}) + \frac{\partial}{\partial y}(\mu\frac{\partial U}{\partial y}) + \frac{\Delta P}{\Delta x} \tag{2}$$

$$\frac{\partial(\varepsilon\rho UV)}{\partial x} + \frac{\partial(\varepsilon\rho VV)}{\partial y} = -\frac{\partial P}{\partial y} + \frac{\partial}{\partial x}(\mu\frac{\partial V}{\partial x}) + \frac{\partial}{\partial y}(\mu\frac{\partial V}{\partial y}) + \frac{\Delta P}{\Delta y} \tag{3}$$

Energy equation:

$$\begin{aligned}
\frac{\partial(\varepsilon\rho UH)}{\partial x} + \frac{\partial(\varepsilon\rho VH)}{\partial y} &= \frac{\partial}{\partial x}(k_{eff}\frac{\partial T_g}{\partial x}) + \frac{\partial}{\partial y}(k_{eff}\frac{\partial T_g}{\partial y}) + \varepsilon\sum_{i=1}^{N}h_i w_i W_i \\
&- \varepsilon\frac{\partial}{\partial x}\left[\rho g\sum_{i=1}^{N}Y_i h_i D_{im}\frac{\partial(\rho g Y_i)}{\partial x}\right] - \varepsilon\frac{\partial}{\partial y}\left[\rho g\sum_{i=1}^{N}Y_i h_i D_{im}\frac{\partial(\rho g Y_i)}{\partial y}\right]
\end{aligned} \tag{4}$$

Species transport equation:

$$\frac{\partial(\varepsilon\rho M_i)}{\partial t} + \frac{\partial(\varepsilon\rho UM_i)}{\partial x} + \frac{\partial(\varepsilon\rho VM_i)}{\partial y} = \frac{\partial}{\partial x}\left[\varepsilon D_{im}\frac{\partial(\rho M_i)}{\partial x}\right] + \frac{\partial}{\partial y}\left[\varepsilon D_{im}\frac{\partial(\rho M_i)}{\partial y}\right] + \varepsilon M_i W_i \tag{5}$$

Ideal gas state equation:

$$P = \rho RT\sum_{n=1}^{\infty}\frac{M_i}{M} \tag{6}$$

### 2.3. Boundary Conditions

Here, the flow pattern in a porous media combustor is determined by the aperture Reynolds number (Re), and the minimum Reynolds number exceeds 500, thus the realizable k-ε turbulence model is selected and the solid-fluid contact walls are the coupling boundary conditions. Inlet velocity boundary conditions are applied, while the outlet is set as the pressure outlet, and the inlet temperature is fixed at 300 K; the outlet gauge pressure is set at 0. To get close to the real situation, the heat transfer condition of the outer wall is set as mixed heat transfer, and the mixed heat transfer of both radiation and convection are considered. The heat transfer expression is:

$$q = h_c(T_W - T_\infty) + \varepsilon\sigma(T_W^4 - T_\infty^4) \tag{7}$$

where $T_w$ and $T_\infty$ represent the wall temperature and environmental temperature (300 K), respectively, and $h_c$ represents the convection heat transfer coefficient of 20 W/(m²·K), while $\varepsilon$ and $\sigma$ represent the wall emissivity and the Stephan–Boltzman constant, respectively

### 2.4. Numerical Simulation Verification

The quality of the grid has a greater impact on the results of the numerical simulation. The micro-burners in this study all use a two-dimensional structure calculation model, the division method of the surface. In this paper, the grid spacing of the micro-burner model is 0.04 mm, and the number of grid units of the heat cycle porous medium hedge burner model is 29,885. The calculation model of the micro-burner is shown in Figure 2a.

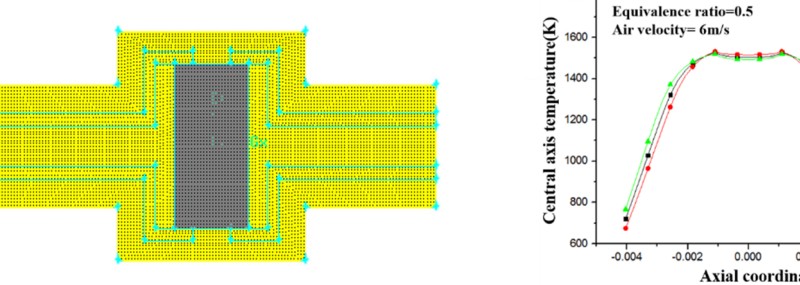

(**a**) Schematic diagram of micro-combustor mesh    (**b**) Grid independence verification

**Figure 2.** Grid units and mesh independence verification.

This study uses the numerical simulation software FLUENT for simulation. Due to the small size of the burner, a temperature equilibrium between the gas phase and the solid phase is assumed in the rapid hedging reaction. The inlet adopts the velocity inlet boundary condition, the outlet is set as the pressure outlet, and the given outlet pressure is atmospheric pressure. In the simulation calculation process, the outer wall and the inner baffle adopt mixed heat transfer boundary conditions. To accelerate the convergence and ensure better stability in the calculation, a high-temperature area "ignition" is set in the porous medium area of the combustion chamber, so that the temperature of the calculation area is set to 1200 K at the initial moment. The fluid density is calculated according to the ideal gas equation, and the gas parameters of the fluid mixture are calculated in the light of the mass-weighted average of the characteristics of the components. The discrete format uses the first-order upwind style, and the single-precision coupled solver is used to solve the governing equations. The convergence of the solution is judged according to the residual value of the governing equation. The energy residual value is set to $10^{-6}$, and the remaining residual values set at $10^{-5}$, while the solution method adopts the SIMPLE algorithm.

Before numerical simulation, the selected numerical method needs to be verified by numerical calculation, including verification of grid independence. The grid independence

verification is mainly to reduce the influence of the grid size on the numerical calculation results, thereby improving the accuracy of the numerical simulation calculation results and reducing the working time of the numerical simulation calculation to ensure the accuracy of the numerical calculation results. The correctness requires a reasonable selection of the grid size. While considering the quality of grid division, the calculation results of the regenerative porous medium hedge burner are verified for grid independence, and the grid sizes are respectively selected as 0.03 mm, 0.04 mm, and 0.05 mm for iterative calculation. The result analysis of the temperature of the central axis of the burner is shown in Figure 2b.

The inlet conditions, boundary conditions and gas parameters of the control burner remain unchanged. When the grid size is selected as 0.03 mm, 0.04 mm, and 0.05 mm, the temperature distribution of the central axis of the burner remains almost unchanged, and the temperature difference is less than 10 K. Therefore, to make the simulation results closer to the actual situation, and considering the number of simulation calculations, it can be determined that in the calculations when the grid spacing is 0.04 mm, the requirements of grid independence in this paper are met.

## 3. Results Discussion and Analysis

### 3.1. Whether the Micro Burner Has a Heat Cycle Channel

When the conventional combustor performs hedging combustion, the methane premixed gas enters the combustion chamber, the two premixed gases oppose each other, and burn in the middle of the combustion chamber and the tail gas exits from both sides of the combustion chamber.

By changing the intake velocity and the equivalent ratio to simulate the methane hedging combustion in the hedged combustor with or without the heat cycle channel, it is found that when the velocity increases or the equivalent ratio increases, the outlet temperature of the combustor without the heat cycle structure is higher than that with the heat cycle structure. As shown in Figure 3a,b. In the two types of burners, the air intake speed is increased from 4 m/s to 8 m/s, the equivalence ratio varies from 0.4 to 0.6, and the convective heat transfer coefficient of the outer wall and the baffle of the burner is set to 20 W/m$^2$·K. Ferroalloy is used as the material of the wall and the partition. In the non- heat cycle combustor, the methane premixed gas enters the combustion chamber and is directly discharged from the combustion chamber, while for the heat cycle burner, when the premixed gas is burned and discharged, the exhaust gas will be preheated in the heat cycle channel. The combustion gas reduces the outlet temperature of the burner. Comparing Figure 3a,b, the intake velocity and equivalence ratio are both increased, and the outlet temperature of the conventional burner s higher than that of the regenerative burner, while both are higher than about 80 K.

Figure 3 shows the relationship between the outlet temperature of the burner and the speed and equivalence ratio. In addition, the study found that as the equivalence ratio increases, the temperature of the center axis of the heat cycle hedge burner is higher than that of the conventional hedge burner, as shown in Figure 4. In the two types of burners, methane is combusted by counter-combustion. The gas from the premix of the conventional burner is burned and discharged directly, and it stays in the combustion chamber for less time, and as the heat loss of the burner is larger, this results in incomplete combustion. However, as the heat cycle burner after the premixed gas is burned, must be discharged through the heat cycle channel, the flue gas can preheat the unburned gas, the combustion enthalpy is greatly increased, the ignition heat of the premixed gas is reduced, the combustion reaction speed is improved, and the heat loss of the burner is reduced. In the heat cycle combustor, the heat cycle channel makes the premixed gas stay longer in the combustion chamber, making the combustion more complete. Therefore, under different equivalence ratios, the temperature of the central axis of the heat cycle combustor is higher than that of the conventional hedge combustor. By changing the equivalent ratio

to simulate methane in the two burners, it is concluded that the CO emissions at the outlet of the heat cycle burner are much lower than that of the conventional burner.

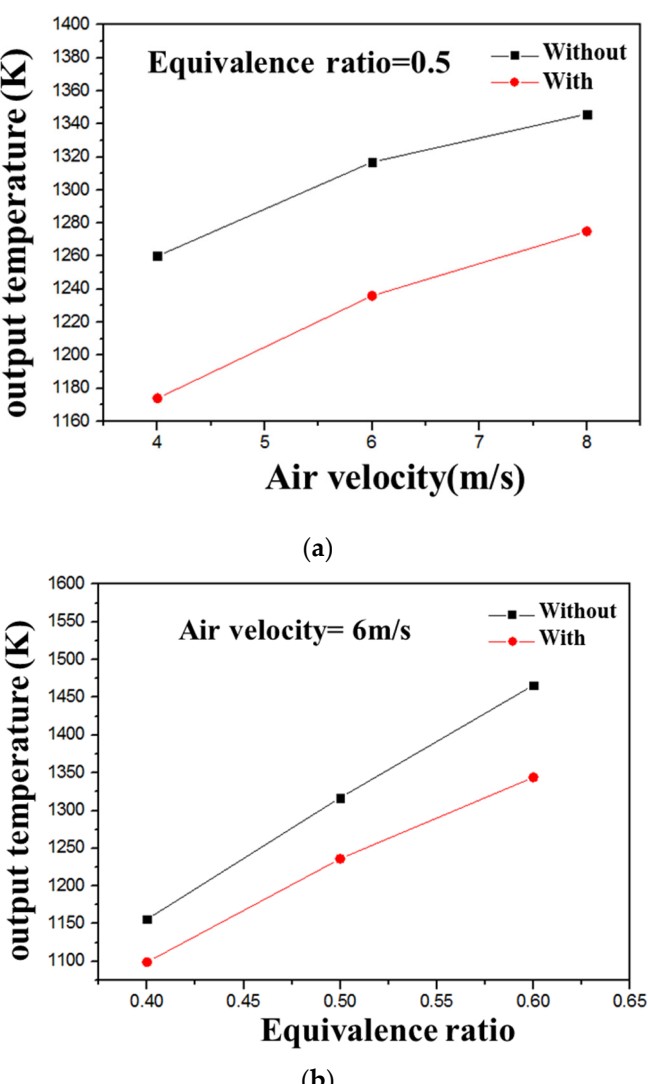

(a)

(b)

**Figure 3.** (**a**) The relationship between outlet temperature and intake speed. (**b**) The relationship between outlet temperature and equivalent ratio.

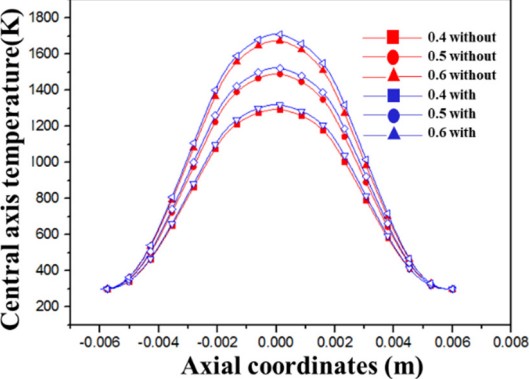

**Figure 4.** The relationship between the core temperature of the combustor with or without heat cycle structure and the equivalent ratio.

### 3.2. Whether the Combustion Chamber Is Filled with Porous Media

Due to the good thermal conductivity of porous media, it can make the heat transfer in the porous media area more intense and the temperature of the porous media area more uniform. To make the temperature distribution in the combustion chamber more uniform, the combustion chamber is filled with porous media. In this paper, $Al_2O_3$ ceramic foam was selected as the porous medium material, and its influence on the temperature distribution of the burner studied.

This paper now looks at the filling of the porous medium $Al_2O_3$ in the combustion chamber. Numerical simulation by changing the flow rate shows that the internal temperature distribution of the combustor filled with porous medium in the combustion chamber is more uniform, while the porous medium filled in the combustion chamber has little effect on the outer wall temperature. The temperature difference of the outer wall surface of the device is about 10 K, as shown in Figure 5. In order to improve the temperature distribution in the combustion chamber, the heat cycle hedging burner is filled with porous media, the thickness of the porous media material is 2 mm, the porosity of the porous media is 0.85, and the PPI is 20. Porous media combustion can be divided into many aspects. In this paper, the inert porous media combustion technology is adopted. There are two different forms of inert porous media combustion. The flame mainly burns at the surface of the porous media or the flame is completely in the combustion process inside the porous medium. Here, the combustion of premixed gas in an inert porous medium is investigated. It can be seen that the methane premixed gas equivalent ratio is 0.4 and 0.5, and filling the porous medium in the combustion chamber makes the radial center temperature rise by about 50 K and become more uniform.

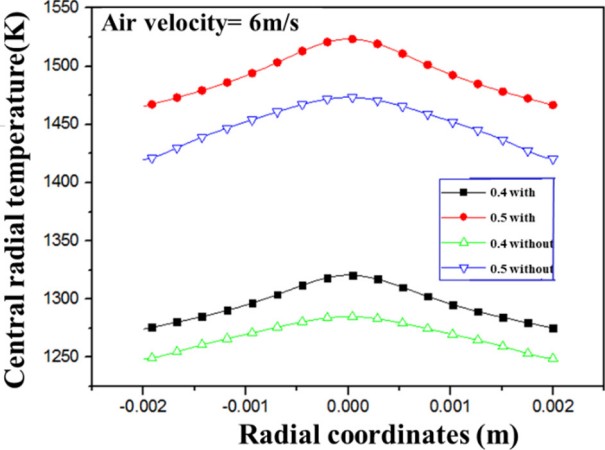

**Figure 5.** Temperature distribution in the radial center of the porous media area.

The temperature cloud diagrams of the two burners with or without porous media when the air flow velocity is 6 m/s and the equivalence ratio is 0.5 are obtained, as shown in Figure 6a,b. It can be seen that when there is no porous medium in the combustion chamber, the area of the high temperature zone in the combustion chamber is relatively small, and when the porous medium is filled, the area of the middle high temperature zone is significantly enlarged. Therefore, filling the porous medium in the combustion chamber can result in the temperature distribution being uniform, enhance the combustion stability, accelerate the combustion rate, and increase the combustion intensity.

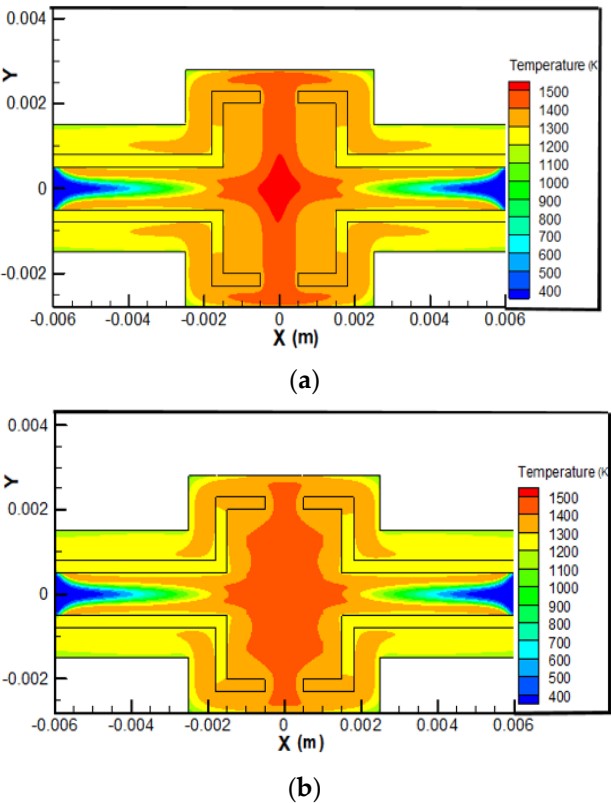

**Figure 6.** Temperature cloud diagram of heat cycle burner. (**a**) Temperature cloud diagram of the porous medium filled in the combustion chamber. (**b**) The temperature cloud diagram of the combustion chamber without porous media.

As shown in Figure 7, When the intake air velocity is in the range of 2 m/s–10 m/s, as the intake velocity increases, the outer wall temperature increases, but for the two combustors with or without porous media, their outer wall temperatures are very close, the difference being about 10 K. Since the porous medium is filled in the combustion chamber without contacting the outer wall surface, and the porous medium is only filled in the middle part of the combustion chamber, not from the inlet of the burner, the porous medium does not play a role in preheating. Therefore, the temperature in the combustion chamber cannot be satisfactorily increased, and the temperature of the outer wall will not increase.

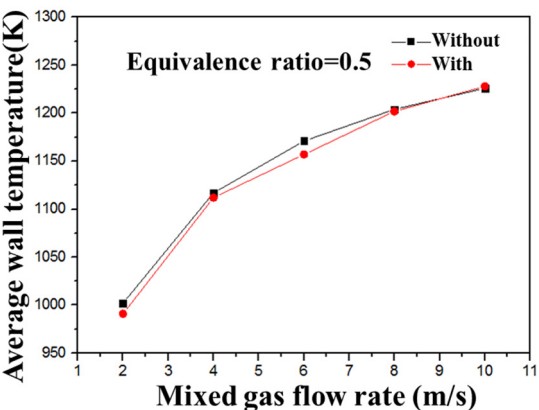

**Figure 7.** Whether the combustion chamber shows a relationship between the average wall temperature of the porous media burner and the intake air flow rate.

### 3.3. Whether There Is a Porous Medium in the Heat Cycle Channel

The heat cycle channel is filled with porous media so that heat energy can be better transferred to the outer wall surface to obtain a higher outer wall surface temperature, thereby improving the heat utilization rate of the burner. First, the influence of intake air flow rate is studied. After simulating two kinds of burners with or without porous media in the regenerative channel by changing the intake air velocity, it was concluded that the center temperature and the external temperature of the two different regenerative burners vary with the intake air flow rate. In the two reactors, the inlet velocity is 5 m/s, 6 m/s, 7 m/s, the equivalent ratio of methane premixed gas is 0.5, and the material of the porous medium filled in the heat cycle channel is $Al_2O_3$ foam ceramic, porous medium The porosity $\varepsilon$ = 0.85, and the PPI is 20. In the two burners, when the equivalence ratio of the premixed gas is 0.5, the relationship between the radial center temperature, the outer wall temperature, and the intake velocity of the two burners under the conditions of different intake flow rates is shown in Figure 8a,b. It can be seen that when the air inlet velocity is 5 m/s, the temperature of the radial center of the regenerative channel without porous medium is higher than that of the porous medium and the velocity is 6 m/s and 7 m/s, the radial center temperature of the two burners is almost the same. This is because when the intake air velocity is 5 m/s, the fuel flow into the combustor is small, and the heat generated by combustion is small, and the porous medium is filled in the heat cycle channel so that the heat transfer between the exhaust gas and the wall is greater than the heat transfer between the exhaust gas and the baffle. Thus, the temperature of the radial center of the heat cycle channel without porous medium is higher than that of the porous medium. When the speed increases, the heat generated by the combustion in the combustion chamber increases, so that the preheating effect of the exhaust gas is similar to the wall heat dissipation effect and the radial center temperature is not much different.

It can be seen from Figure 8b that with the increase of the intake air velocity, the temperature of the outer wall surface of the combustor filled with porous medium in the heat cycle channel is significantly higher than that of the combustor without porous medium. The average temperature difference of the outer wall surface is 106 K at a speed of 4 m/s, and the temperature difference is 116 K when the speed is 8 m/s. With the increase of the intake speed, the difference of the outer wall surface of the two burners is about 110 K. It shows that filling the porous medium in the heat cycle channel is beneficial to obtain a higher outer wall temperature, which is in line with the requirements of the micro-thermal photoelectric system. This is because the exhaust gas generated in the combustion chamber must enter the heat cycle channel, and must contact the outer wall surface and transfer heat to the outer wall surface through heat conduction, heat convection, and thermal radiation. When the heat cycle channel is filled with porous medium, the porous medium has better heat conduction and heat storage effect, so that the heat loss of the outlet flue gas is less, so that the temperature of the outer wall surface is higher. Therefore, it is more suitable for the micro-thermal photovoltaic system to fill the porous medium in the heat cycle channel.

Through numerical simulation, the relationship between the temperature distribution of the combustor filled with porous media in both the combustion chamber and the heat cycle channel and the intake air flow rate is obtained; the control equivalence ratio is 0.5, as shown in Figure 9a,b. As the inlet velocity of methane premixed gas increases, the combustion characteristics in the combustor are also different. It can be seen from Figure 9a that as the speed increases, the maximum temperature in the combustion chamber hardly changes and is maintained at about 1500 K. However, when the speed is 5 m/s, the highest temperature appears at the point where the intake pipe enters the combustion chamber. When the speed increases, the highest temperature will gradually move to the middle of the combustion chamber. When the velocity increases, the highest temperature appears in the combustion chamber center, this is because in the center of the combustion chamber, the two airflows oppose each other, and the increase in turbulence makes the combustion more intense. However, it can be seen from Figure 8a that as the intake speed increases, the radial center temperature of the combustion chamber increases. In addition, it can be seen

from Figure 9b that since the center of the outer wall is the outlet of the combustion exhaust gas from the combustion chamber, the temperature of the outer wall is the highest at the center, and gradually decreases at both sides, as the speed increases, and the fuel flow into the combustion chamber also continues to increase. Therefore, the heat released in the combustion chamber also increases, so the wall temperature increases with the increase in speed.

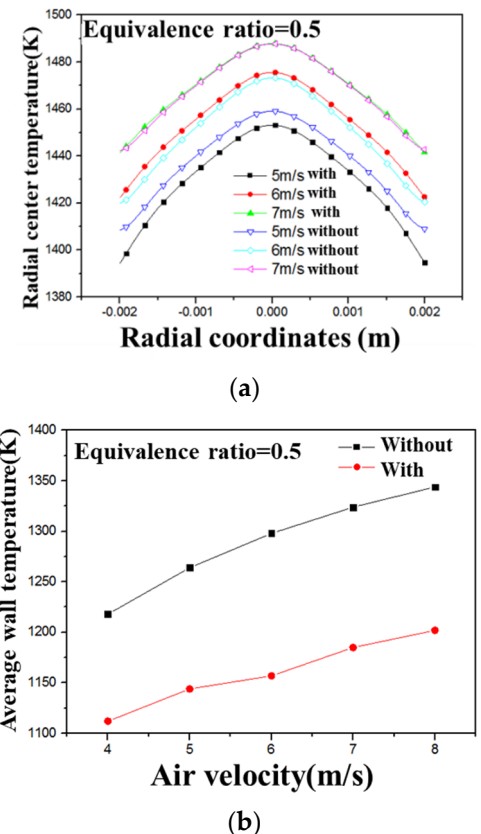

(**a**)

(**b**)

**Figure 8.** The relationship between the temperature distribution and speed of the burner. (**a**) The relationship between radial center temperature and intake speed. (**b**) The relationship between average wall temperature and intake speed.

In addition, the effect of equivalence ratio is studied here. This section controls the equivalent ratio for numerical simulation to study the relationship between the central axis temperature and the wall temperature of the heat cycle porous media burner and the equivalent ratio. The methane premixed gas flow rate of the burner is set to 6 m/s and the initial temperature of the premixed gas to 300 K, the material of the outer wall surface of the burner and the inner baffle plate is selected as iron-chromium alloy, then the definition formula of equivalent ratio is as follows:

$$\varnothing = \frac{(A/F)}{(A_0/F_0)} \tag{8}$$

$A_0/F_0$ represents the ratio of air to fuel in the actual combustible mixture, $A/F$ represents the ratio of air to fuel when the fuel in the combustible mixture is theoretically completely burned.

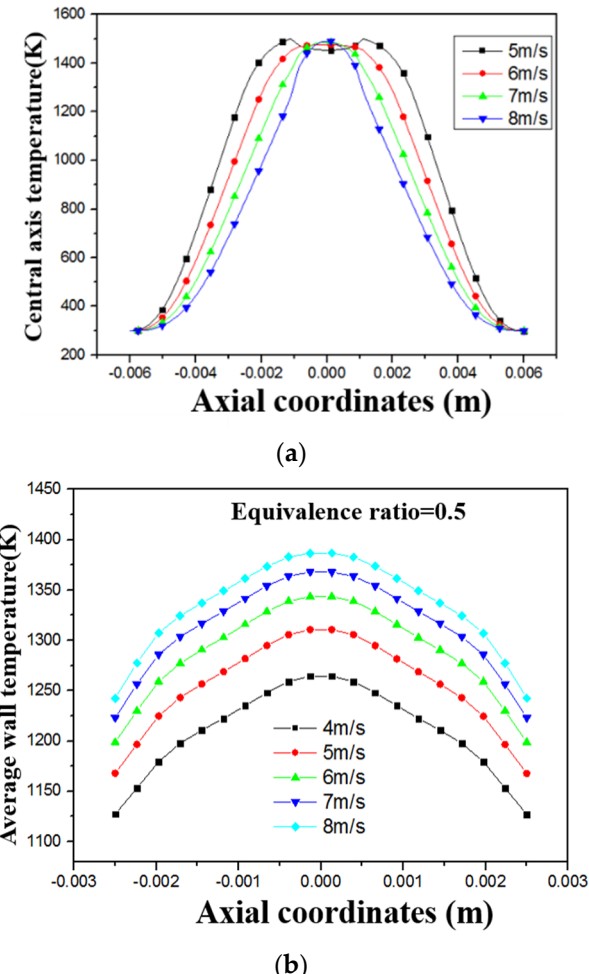

**Figure 9.** The relationship between temperature distribution and the velocity of the porous media burner. (**a**) The relationship between burner core temperature and speed. (**b**) The relationship between the temperature of the outer wall of the burner and the speed.

As shown in Figure 10, when methane is combusted in a heat cycle porous medium combustor, as the equivalence ratio increases from 0.3 to 0.9, the temperature of the central axis of the combustor increases with the increase in equivalence ratio. When $\varnothing < 1$, it is in the lean burn zone, and the variation range of the equivalence ratio is 0.3–0.9, which is all in the lean burn zone. Therefore, under these equivalence ratios, the conversion rate of methane is high, basically all is burnt completely. When the equivalence ratio $\varnothing = 0.3$, the maximum temperature of the combustion chamber is 1074 K. When the equivalence ratio $\varnothing = 0.9$, the maximum temperature of the combustion chamber is 2100 K. When the equivalence ratio increases from 0.3 to 0.9, the maximum temperature of the combustion chamber increases by 1026 K. This is because the equivalence ratio of the premixed gas increases, and the mass fraction of methane gas in the premixed gas increases. That is, the content of methane gas of the same volume of premixed gas is passed under the condition of keeping the inlet flow rate constant. The higher the temperature, the higher the heat released by combustion, so the maximum temperature in the combustion chamber is constantly increasing. When $\varnothing > 1$, it is in the rich combustion zone. When the equivalence ratio $\varnothing = 1.1$, the increase in the temperature at the center of the combustion chamber is smaller than that in the lean zone. This is because when in the rich combustion zone, the mass fraction of methane in the premixed gas is large, the premixed gas is not fully burned in the combustion chamber, resulting in less heat released.

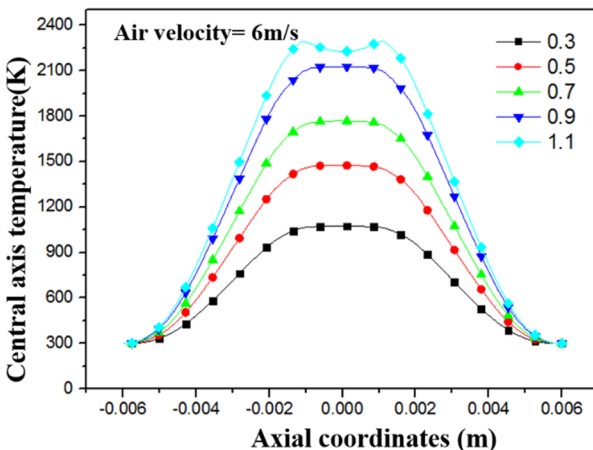

**Figure 10.** The relationship between equivalence ratio and core temperature.

The effect of the same equivalence ratio on the temperature distribution of the outer wall of the combustor is also not negligible. Figure 11 shows the effect of the equivalence ratio on the temperature of the outer wall of the heat cycle porous media burner. As the mass fraction of methane increases, the combustion in the combustion chamber releases more heat and the temperature in the combustion chamber increases. The temperature of the combustion exhaust gas rises, so that the temperature of the outer wall surface rises. The equivalence ratio increases from 0.3 to 0.4, and the outer wall surface temperature rises by 160 K. When it rises from 0.4 to 0.5, the outer wall temperature rises by about 110 K. From 0.6 to 0.7, the temperature of the outer wall only increased by 60 K, and compared with the temperature distribution of the outer wall of the heat cycle channel without porous media, its high temperature zone was mainly concentrated in the middle part of the burner, while the temperature difference between the two sides and the middle was large. The temperature distribution on the outer wall surface of the porous medium is relatively uniform, and the temperature gradient along the wall surface is small, which avoids excessive temperature concentration. This shows that as the equivalence ratio increases, the temperature of the outer wall surface increases, and the equivalence ratio is smaller and the outer wall temperature rises more.

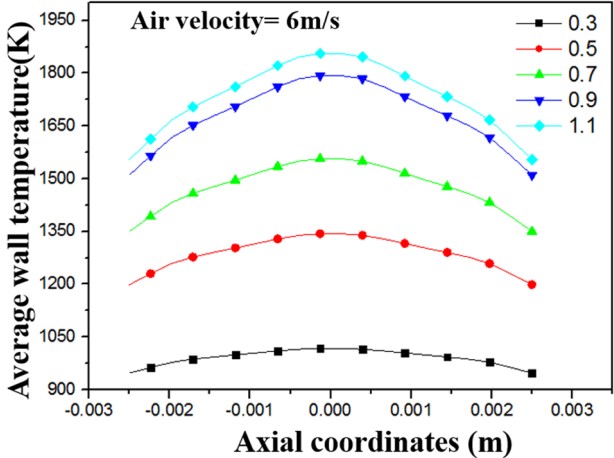

**Figure 11.** The relationship between the outer wall temperature and the equivalent ratio.

## 4. Conclusions

In this paper a micro-burner was designed based on the micro-thermal photoelectric system, and the performance of the micro-burner improved by increasing the heat cycle structure and filling the porous medium. The combustion characteristics in the heat cycle

porous medium combustor were studied by numerical simulation, and the influence of operating conditions such as the distribution of the porous medium $Al_2O_3$ in the combustor, fuel flow rate, and equivalent ratio on the outer wall temperature and core temperature of the combustor investigated. The study found that when the porous medium is filled in the combustion chamber, the temperature distribution in the combustion chamber is more uniform because porous medium has good heat conduction and heat storage capabilities. When the porous medium is filled in the heat cycle channel, the temperature of the outer wall surface of the burner increases, and the temperature of the outer wall surface becomes more uniform. In a heat cycle porous medium combustor, with the increase of the intake air flow rate, the temperature of the outer wall of the combustor gradually increases, and the maximum temperature in the combustion chamber does not change much. With the increase of the equivalence ratio, the temperature of the outer wall of the combustor when $\varnothing > 1$, is in the rich combustion zone and methane combustion is incomplete. The research provides a theoretical basis for the further design of high-efficiency combustion micro-scale burners.

**Author Contributions:** Writing—original draft preparation, F.W.; conceptualization, S.F.; writing—review and editing, F.W. and S.F.; supervision, X.L. and Y.Y.; funding acquisition, Y.Y. All authors have read and agreed to the published version of manuscript.

**Funding:** The authors gratefully acknowledge financial support from the Chongqing Special Equipment Inspection and Research Institute through the funds (No. CQTJKY202101) and Chongqing Administration for Market Regulation through the funds (No. CQZJKY2018023) and the Graduate Scientific Research and Innovation Foundation of Chongqing (CYS21018).

**Institutional Review Board Statement:** Not applicable.

**Informed Consent Statement:** Not applicable.

**Data Availability Statement:** Not applicable.

**Conflicts of Interest:** We declare that we have no financial and personal relationships with other people or organizations that can inappropriately influence our work. There is no professional or other personal interest of any nature or kind in any product, service and/or company that could be construed as influencing the position presented in, or the review of the manuscript entitled, "Numerical Study on the Characteristics of Methane Hedging Combustion in a Heat Cycle Porous Media Burner".

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
