# Peer review of "Numerical Study on the Characteristics of Methane Hedging Combustion in a Heat Cycle Porous Media Burner"

_processes, doi:10.3390/pr9101733_

Round 1

Reviewer 1 Report

Dear authors, the paper presents really interested result.

The only thing is a need to present more details regarding description of numerical analysis: clear definition and Figure with boundary conditions, Figure with meshes used in simulations, common table with all submodels and input data (turbulent flow, heat transfer, chemistry etc.)

Author Response

Reviewer: 1

   The only thing is a need to present more details regarding description of numerical analysis: clear definition and Figure with boundary conditions, Figure with meshes used in simulations, common table with all sub models and input data (turbulent flow, heat transfer, chemistry etc.)

Response: We sincerely appreciate your comments and valuable suggestions. We have studied the comments carefully and have made corrections accordingly. We have added and described the Figure with boundary conditions, Figure with meshes used in simulations, common table with all sub models and input data in the manuscript. The revised parts in the manuscript are marked in red, included line 217 to line 236.

Reviewer 2 Report

The Ronney burner is called a "Swiss roll" not "Swiss bagel" - page 2.

The dimensions in Figure 1 are too small to read.

Page 5: I'm not sure what "normal pressure" at the outlet is. Do you mean atmospheric pressure?

Figure 2 (or text around it): Can you specify the number of cells/grid points you used in the calculations?

Figures 3 and others: Use a more descriptive language in the legends. Something besides "no" and "have" to describe the inclusion of the heat cycle channel.

Check the English and grammar throughout. Also, there are some stray word phrases (e.g., figures) throughout the manuscript that need to be removed or corrected.

Author Response

Reviewer: 2

1.The Ronney burner is called a "Swiss roll" not "Swiss bagel" - page 2.

Response: We sincerely appreciate your comments and valuable suggestions. We have revised Swiss bagel to the Swiss roll in the manuscript, in line124 to 128.

2.The dimensions in Figure 1 are too small to read.

Response: Thank for your suggestions. we have revised Figure 1 again in the manuscript, in line192 to 194.

3.Page 5: I'm not sure what "normal pressure" at the outlet is. Do you mean atmospheric pressure?

Response: Thank for your recommendation. The we mentioned that normal pressure at the outlet refers to atmospheric pressure in the manuscript, and we have revised it, in line239 to 241.

4.Figure 2 (or text around it): Can you specify the number of cells/grid points you used in the calculations?

Response: We sincerely appreciate your comments. We have added the number of grid points used in the calculations in the manuscript, in line228 to line 236.

5.Figures 3 and others: Use a more descriptive language in the legends. Something besides "no" and "have" to describe the inclusion of the heat cycle channel.

Response: Thank for your suggestions. We have modified "no" and "have" in all legends to “with” and “without” to describe the inclusion of the heat cycle channel, which will be more appropriate.

6.Check the English and grammar throughout. Also, there are some stray word phrases (e.g., figures) throughout the manuscript that need to be removed or corrected.

Response: Thank for your recommendation. We have revised the stray word phrases and wrong words throughout the manuscript.